# Systemic immune inflammation index as a predictor of disease severity in tetanus patients: A retrospective observational study

**Dai Cheng**[1], **Ding Wenying**[2], **Huang Jizheng**[2]*, **Sun Wei**[1], **Li Liang**[1], **Han Guolei**[1], **Yang Hao**[1]

1 Department of Intensive Care Medicine, No.2 People's Hospital of Fuyang City, Fuyang, China,
2 Department of Hospital Infection Management, No.2 People's Hospital of Fuyang City, Fuyang, China

* hjzh_ly@126.com

**Data Availability Statement:** The data that support the findings of this study are not openly available due to reasons of sensitivity and are available from the Ethics Committee of No.2 People's Hospital of

## Abstract

### Objective

This study aimed to analyze the predictive value of the systemic immune inflammation index (SII) for the severity of disease in tetanus patients.

### Methods

Clinical data of 34 tetanus patients admitted to the Second People's Hospital of Fuyang from **January 1, 2019 to December 31, 2022** were analyzed. Based on whether patients received intensive care unit (ICU) treatment after admission, the patients were divided into ICU and non-ICU groups. The diagnostic value of SII for the severity of tetanus was assessed.

### Results

Among the 34 patients, 18 (52.90%) were classified into the ICU group, and 16 (47.10%) into the non-ICU group. There were statistically significant differences in white blood cell count, platelet count, neutrophil count, and SII between the two groups (P<0.05). Logistic regression analysis revealed that SII was a risk factor for tetanus patients requiring ICU treatment. The area under the curve (AUC) for SII predicting ICU treatment in tetanus patients was 0.896 (95% CI 0.790–1.000, P<0.001).

### Conclusion

The SII can serve as an objective predictive indicator for tetanus patients requiring ICU treatment.

## Introduction

Tetanus is an infectious disease caused by Clostridium tetani, characterized by persistent, tonic contractions of skeletal muscles and paroxysmal spasms in patients [1,2]. Due to the

Fuyang City (fyhmf@163.com), upon reasonable request.

**Funding:** The author(s) received no specific funding for this work.

**Competing interests:** The authors have declared that no competing interests exist.

nonspecific early clinical symptoms of tetanus, patients are typically initially admitted to general wards, and transfer to the intensive care unit (ICU) for treatment occurs when the condition becomes severe, leading to delayed intervention and posing a threat to patients' lives. Therefore, guidelines emphasize the need for more graded schemes and clinically relevant studies on prognosis [2,3].

The Systemic Immune Inflammation Index (SII) is a novel immunoinflammatory marker that integrates peripheral blood platelet, neutrophil, and lymphocyte counts. It is considered an effective indicator reflecting the systemic immune-inflammatory status and prognosis [4]. Studies have shown that SII has excellent clinical value in assessing the prognosis of patients with rheumatoid arthritis, cancer, infectious diseases, and other conditions [5–8]. A study has revealed that the SII in severe COVID-19 patients is significantly elevated compared to non-severe cases, demonstrating a substantial predictive value of SII for assessing the severity of the patients' condition [9]. However, there are currently few reports on the relationship between SII and the occurrence and severity of tetanus. This study retrospectively analyzed the clinical treatment of 34 tetanus patients, exploring the clinical value of SII in predicting the severity of disease in tetanus patients. The findings are reported as follows.

## 2 Materials and methods

### 2.1 Study design and patients

The studies involving human participants were reviewed and approved by Ethics Committee of No.2 People's Hospital of Fuyang City(20231210056).The patients/participants provided their written informed consent to participate in this study. All methods were implemented in accordance with relevant guidelines and regulations.The study was conducted in accordance with the Helsinki Declaration of 1975. The diagnosis of tetanus met the WHO definition of tetanus.

We reviewed the hospital database between December 5th, 2023 and December 31st, 2023 that involved adult tetanus patients who were admitted to the No.2 People's Hospital of Fuyang City between January 1, 2019 to December 31, 2022. Inclusion criteria were as follows: ① Patients clinically diagnosed with tetanus; ② Age>18 years old; ③ Complete clinical data. Exclusion criteria were: ① Discharge or death within 24 hours of admission; ② Incomplete clinical data. A total of 34 patients were included, comprising 22 males (64.70%) and 12 females (35.30%), with an average age of (62.06±12.88) years.

### 2.2 Data collection

We collected clinical data from 34 tetanus patients and divided them into ICU and non ICU groups based on whether they received ICU treatment.Data recorded included patient gender, age, underlying diseases, Ablett classification, time from onset to medical treatment, total hospital stay, white blood cell count, platelet count, total bilirubin, creatinine, creatine kinase, creatine kinase isoenzyme, neutrophil count, lymphocyte count, and the Systemic Immune Inflammation Index (SII) calculated based on blood routine results. Laboratory indicators are tested within 24 hours of patient admission.SII = Neutrophil count ($\times 10^9$/L) $\times$ Platelet count ($\times 10^9$/L) / Lymphocyte count ($\times 10^9$/L) [4].

### 2.3 Treatment

Treatment of non-ICU patients: Patients were isolated in a single room to reduce stimulation such as sound and light, wound treatment, diazepam was used to control spasms, neutralize free toxins (tetanus antitoxin), eliminate Bacillus (penicillin + tinidazole), etc.

Treatment of patients in ICU group: The patient is isolated in a single room to avoid stimulation such as sound and light, wound treatment, neutralization of free toxins (tetanus antitoxin), elimination of Bacillus (penicillin + tinidazole), mechanical ventilation, long-term mechanical ventilation requires tracheotomy, mechanical ventilation analgesia, sedation and control of spasticity, nutritional support and other symptomatic treatment.

## 2.4 Statistical analysis

Statistical analysis was performed using SPSS 25.0. Continuous data were expressed as "$\bar{x}\pm s$" and analyzed using independent sample t-tests. Categorical data were presented as case numbers (n) and percentages (%) and analyzed using $\chi^2$ tests. Binary logistic regression analysis was used for multifactor analysis. Receiver Operating Characteristic (ROC) curves were drawn to analyze the diagnostic value of SII for the severity of tetanus, with the cutoff value determined as the point with the maximum Youden index. A $p$ value $<0.05$ wasconsidered to be statistically significant.

# 3 Results

## 3.1 Univariate analysis of tetanus patients receiving ICU treatment

During the study period, a total of 41 tetanus patients were admitted to the Second People's Hospital of Fuyang. Among them, 6 patients were excluded due to insufficient clinical data, and 1 patient was excluded due to discharge within 24 hours after admission. Finally, 34 tetanus patients were included as the study objects. Among the 34 patients, 18 (52.90%) were categorized into the ICU group, and 16 (47.10%) into the non-ICU group based on whether they received ICU treatment. There were no statistically significant differences between the two groups in terms of age, gender, underlying diseases, time from onset to medical treatment, total bilirubin, creatinine, creatine kinase, creatine kinase isoenzyme, and lymphocyte count ($P>0.05$). However, statistically significant differences were observed in white blood cell count, platelet count, neutrophil count, and the Systemic Immune Inflammation Index (SII) between the two groups ($P<0.05$). Refer to Table 1 for details.

## 3.2 Multivariate analysis of tetanus patients receiving ICU treatment

Building upon the results of univariate analysis, variables with statistically significant differences, including white blood cell count, platelet count, neutrophil count, and the Systemic Immune Inflammation Index (SII), were selected as independent variables. Binary logistic regression analysis was conducted with the acceptance of ICU treatment as the dependent variable. The results indicated that SII is a risk factor for tetanus patients requiring ICU treatment. Refer to Table 2 for details.

## 3.3 Predictive value of SII for tetanus patients receiving ICU treatment

The SII demonstrated significant predictive value for tetanus patients requiring ICU treatment, with an area under the curve (AUC) of 0.896 (95% CI 0.790–1.000, $P<0.001$). At the optimal cutoff value of 892.98, the sensitivity of SII in predicting tetanus patients requiring ICU treatment was 83.3%, with a specificity of 87.5%. Refer to Fig 1 for the ROC curve illustrating the predictive performance of SII. Refer to Fig 1 for details.

## 3.4 Treatment

34 patients with tetanus were all treated with antibiotics, using metronidazole and penicillin. 72.22% (13/18) of the ICU group received TAT treatment, while 56.25% (9/16) of the non-

**Table 1. Univariate analysis of tetanus patients receiving ICU treatment.**

| Variable | | ICU Group (n = 18) | Non-ICU Group (n = 16) | $\chi^2$/ t Value | P Value |
|---|---|---|---|---|---|
| Gender [n(%)] | | | | 0.216 | 0.642 |
| | Male | 11 (61.11) | 11 (68.75) | | |
| | Female | 7 (38.89) | 5 (31.25) | | |
| Underlying Diseases [n(%)] | | | | 2.862 | 0.091 |
| | Present | 4 (22.22) | 8 (50.00) | | |
| | Absent | 14 (77.78) | 8 (50.00) | | |
| Age (years) | | 62.22±12.55 | 61.88±13.65 | 0.077 | 0.939 |
| Time from Onset to Treatment (days) | | 2.78±1.66 | 3.88±3.59 | -1.164 | 0.253 |
| White Blood Cell Count ($\times10^9$/L) | | 10.00±3.14 | 6.55±1.99 | 3.776 | 0.001 |
| Platelet Count ($\times10^9$/L) | | 284.28±81.61 | 219.13±68.19 | 2.508 | 0.017 |
| Total Bilirubin ($\times10^9$/L) | | 15.33±5.36 | 16.37±9.54 | -0.398 | 0.693 |
| Creatinine ($\times10^9$/L) | | 62.39±11.88 | 63.94±11.11 | -0.393 | 0.698 |
| Creatine Kinase ($\times10^9$/L) | | 334.33±205.73 | 222.75±207.19 | 1.573 | 0.125 |
| Creatine Kinase Isoenzyme ($\times10^9$/L) | | 21.11±10.77 | 18.50±16.33 | 0.556 | 0.582 |
| Neutrophil Count ($\times10^9$/L) | | 8.16±3.36 | 4.27±1.84 | 4.114 | <0.001 |
| Lymphocyte Count ($\times10^9$/L) | | 1.45±0.78 | 1.61±0.58 | -0.649 | 0.521 |
| SII | | 2222.15±2197.33 | 624.84±349.71 | 3.041 | 0.007 |

ICU group received TAT treatment. There was no statistically significant difference between the two groups ($\chi^2$ = 0.946, P = 0.331).100.00% (18/18) of ICU group patients used spasm control drugs, while 68.75% (11/16) of non-ICU group patients used spasm control drugs. The difference between the two groups was statistically significant ($\chi^2$ = 0.946, P = 0.331). 100.00% (18/18) of ICU patients used sedation and mechanical ventilation, while 0% (0/16) of non-ICU patients used sedation and mechanical ventilation, with a statistically significant difference between the two groups ($\chi^2$ = 34.000, P<0.001). Refer to Table 3 for details.

## 3.5 Prognosis

The prognosis of tetanus patients varied based on the initial treatment location. Among those initially treated in the ICU, 6 patients were admitted, 5 were cured, and 1 patient opted for treatment discontinuation. For patients transferred from general wards to the ICU, 12 were admitted, 9 were cured, and 3 patients chose to discontinue treatment. One patient initially admitted to a general ward, who deteriorated but refused transfer to the ICU, opted for treatment discontinuation. All 15 patients continuously treated in general wards were cured. Refer to Table 3 for a comparison of the cure rates among different treatment groups. Refer to Table 4 for details.

**Table 2. Multivariate analysis of tetanus patients receiving ICU treatment.**

| Variable | β | Wald | P Value | OR | 95% C.I.for OR | |
|---|---|---|---|---|---|---|
| | | | | | Lower | Upper |
| SII | 0.003 | 4.029 | 0.045 | 1.003 | 1.000 | 1.006 |
| Neutrophil Count | 0.673 | 3.399 | 0.065 | 1.961 | 0.958 | 4.010 |
| Constant | -6.844 | 6.282 | 0.012 | 0.001 | | |

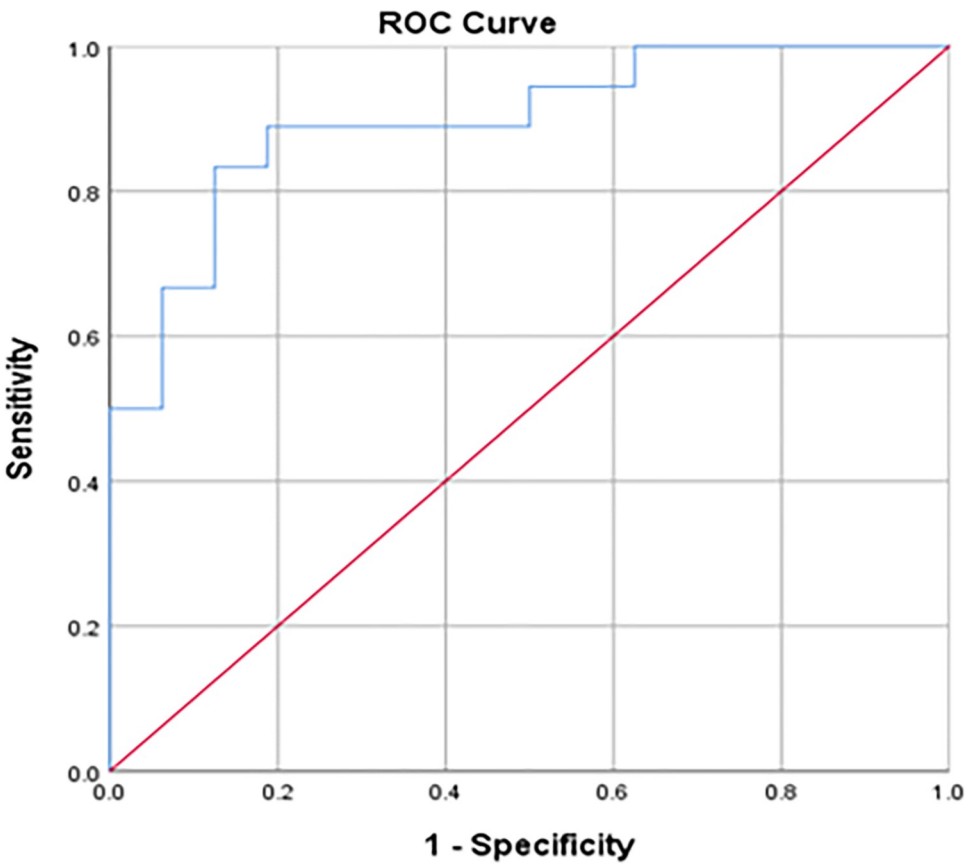

**Fig 1. ROC curve of SII for predicting ICU treatment in tetanus patients.**

**Table 3. Comparison of treatment between ICU and non-ICU patients.**

| Variable | | ICU Group (n = 18) | Non-ICU Group (n = 16) | χ²/ t Value | P Value |
|---|---|---|---|---|---|
| Using TAT | | | | 0.946 | 0.331 |
| | Yes | 13 | 9 | | |
| | No | 5 | 7 | | |
| Using spasm control drugs | | | | - | 0.016* |
| | Yes | 18 | 11 | | |
| | No | 0 | 5 | | |
| Using sedatives | | | | 34.000 | <0.001 |
| | Yes | 18 | 0 | | |
| | No | 0 | 16 | | |
| receiving mechanical ventilation | | | | 34.000 | <0.001 |
| | Yes | 18 | 0 | | |
| | No | 0 | 16 | | |

* Fisher's exact test method.

**Table 4. Comparison of cure rates in different initial treatment settings.**

| Group | Total Number (cases) | Discontinuation/Death (cases) | Cured (cases) | Cure Rate (%) |
|---|---|---|---|---|
| First ICU Treatment | 6 | 1/0 | 5 | 83.33 |
| Transfer to ICU Treatment | 12 | 3/0 | 9 | 75.00 |
| Continuous General Ward Treatment | 15 | 0/0 | 15 | 100.00 |
| Total | 33 | 4/0 | 29 | 87.88 |

## 4 Discussion

Tetanus is a vaccine-preventable disease that remains prevalent in many low- and middle-income countries, posing ongoing challenges [10,11]. Studies suggest that early initiation of intensive care and nursing for severe and very severe (III/IV grade) patients can reduce the mortality rate among tetanus patients [12,13]. Therefore, predicting the severity of tetanus is crucial for patient treatment and prognosis.

The main principles of tetanus treatment include: sedation, analgesia, muscle relaxation to control spasms, correction of autonomic dysfunction to avoid exhaustion; Thorough debridement and anti tetanus clostridium treatment; Neutralizing toxins in the circulatory system; symptomatic and supportive treatment [14]. Some studies have pointed out that airway management is the key to reducing the risk of death. For severe and extra severe patients, early tracheotomy and admission to intensive care units with mechanical ventilation support are recommended for treatment [15]. Therefore, early identification of severe and extra severe patients and early transfer to the ICU for treatment are necessary for the successful treatment of patients.

Currently, the assessment of tetanus severity often relies on the Ablett grading system [2,16]. In this study, 13 patients initially admitted to general wards based on Ablett grading later required transfer to the ICU as their conditions worsened. Some patients exhibited mild symptoms upon admission and deteriorated after being admitted to general wards, leading to delayed ICU treatment. Cai Miaotian et al. [17] highlighted the limitations of the Ablett grading system, emphasizing its subjective nature and lack of quantitative precision. In contrast, the Systemic Immune Inflammation Index (SII) can predict the severity of the disease upon admission, guiding ICU admission and allowing for early intervention to prevent progression to severe conditions. The results of this study demonstrated that the SII was significantly higher in tetanus patients undergoing ICU treatment compared to those not requiring ICU, and that an increase in SII was a risk factor leading to ICU admission for tetanus patients, suggesting that SII could reflect the severity of illness in tetanus patients. In this study, the AUC of the ROC curve for SII predicting ICU treatment in tetanus patients was 0.896, demonstrating high sensitivity and specificity. This shows that the measurement of SII is conducive to early identification of severe tetanus patients by clinicians, so that more stringent medical supervision and more active treatment measures can be taken for patients with severe tetanus to improve patient prognosis.

In summary, the SII is a convenient and cost-effective indicator that can serve as an objective predictor for tetanus patients requiring ICU treatment. However, this study is retrospective in nature, which inherently entails drawbacks such as potential data bias, numerous confounding factors, recall bias, and difficulties in ensuring data quality and completeness. In future studies, we will conduct a multi-center, prospective study with an expanded sample size to validate and refine the findings of this research, thereby enhancing its clinical application value. Additionally, the lack of dynamic monitoring of SII changes during hospitalization and the absence of an in-depth exploration of the correlation between SII changes and the severity of the patient's condition require further investigation in future studies.

## Author Contributions

**Conceptualization:** Huang Jizheng.

**Data curation:** Sun Wei, Li Liang, Han Guolei, Yang Hao.

**Formal analysis:** Sun Wei.

**Investigation:** Ding Wenying, Li Liang, Han Guolei.

**Methodology:** Huang Jizheng.

**Software:** Yang Hao.

**Supervision:** Huang Jizheng.

**Writing – original draft:** Dai Cheng.

**Writing – review & editing:** Huang Jizheng.

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
