## [Decision Letter · Decision Letter 0]

27 Aug 2024

PONE-D-24-16126Systemic Immune Inflammation Index as a Predictor of Disease Severity in Tetanus Patients: A retrospective Observational StudyPLOS ONE

Dear Dr. HUANG,

Thank you for submitting your manuscript to PLOS ONE. After careful consideration, we feel that it has merit but does not fully meet PLOS ONE’s publication criteria as it currently stands. Therefore, we invite you to submit a revised version of the manuscript that addresses the points raised during the review process.

**What was the total number of tetanus patients seen in the hospital during the study period, and what was the number of those that were excluded due to insufficient data, those excluded on the ground that they were discharged within 24 hours of admission, and those excluded because they were below the study adult age range.**

**The study was conducted in adults; however, the age range of adults was not stated. This should be part of the inclusion criteria.**

**The study's retrospective design, which limits control over selection bias and missing data, should be considered a drawback in addition to those mentioned in the discussion section. The recommendation for a much larger study is in order; however, this preferably should be a prospective study.**

**The references need to be reviewed; most do not conform to PLOS One reference guideline (https://journals.plos.org/plosone/s/submission-guidelines). Furthermore, I could not find references 6, 15 and 16 from literature search.**

**A majority of the References were more than five years; ideally, at least 60-70% of the References should be less than 5 years post publication. This must be addressed.**

We look forward to receiving your revised manuscript.

Kind regards,

Innocent Ijezie Chukwuonye, MBBS, FMCP(Internal Medicine)

Academic Editor

PLOS ONE

Journal Requirements:

2. In the online submission form, you indicated that The data underlying the results presented in the study are available from corresponding author: Huang Jizheng, hjzh_ly@126.com

Reviewers' comments:

Reviewer's Responses to Questions

**Comments to the Author**

1. Is the manuscript technically sound, and do the data support the conclusions?

Reviewer #1: Partly

Reviewer #2: Yes

2. Has the statistical analysis been performed appropriately and rigorously? 

Reviewer #1: Yes

Reviewer #2: Yes

3. Have the authors made all data underlying the findings in their manuscript fully available?

Reviewer #1: Yes

Reviewer #2: Yes

4. Is the manuscript presented in an intelligible fashion and written in standard English?

Reviewer #1: Yes

Reviewer #2: Yes

5. Review Comments to the Author

**Reviewer #1:** I would like to thank the author(s) for the contribution and their manuscript. I have to highlight few comments to further improve this manuscript.

1- There are few scattered spelling mistakes like the word "conformed" in Page 9 and the word "tracheotomy" in Page 18.

2- The author highlighted that the total sample size was 34. However, what was the sample size before the exclusion criteria, in other words, how many patients were admitted with tetanus?

3- The method section seems a bit confusing and not reproducible. The author stated that the patients were categorised into ICU and non-ICU group. The word "categorised" seems inappropriate in this context as patients were not randomised. [This categorisation introduces bias]

4- The author need to illustrate when was the SII calculated. Was it calculated upon admission? Also, is there a role of monitoring serial or daily SII to predict progression of severity and/or morbidity?

5- Was the treatment similar to both arms of the study (the ICU and non-ICU groups)?

6- The discussion section highlights the importance of timely management. However, it does not answer the topic clearly which is the role of SII as this is the cornerstone of this manuscript and their is lack of literature review.

**Reviewer #2:** You should follow journal guide.

Unistyle references.

Add more figure to clarify your work.

Well done.

You should follow journal guide.

Unistyle references.

Add more figure to clarify your work.

Well done.

6. PLOS authors have the option to publish the peer review history of their article (what does this mean?). If published, this will include your full peer review and any attached files.

Reviewer #1: No

Reviewer #2: **Yes: **Yahya Ali Abdulkareem Abodea

---

## [Author Response · Author response to Decision Letter 0]

1 Dec 2024

Dear Academic Editor and reviewers,

Thank you very much for your kindly comments on our manuscript.There is no doubt that these comments are valuable and very helpful for revising and improving our manuscript. In what follows, we would like to answer the questions you mentioned and give detailed account of the changes made to the original manuscript.

Academic Editor:

1.Comment:What was the total number of tetanus patients seen in the hospital during the study period, and what was the number of those that were excluded due to insufficient data,those excluded on the ground that they were discharged within 24 hours of admission, and those excluded because they were below the study adult age range.

Reply:In the results section, corresponding explanations have been added and highlighted in red font.“During the study period, a total of 41 tetanus patients were admitted to the Second People's Hospital of Fuyang. Among them, 6 patients were excluded due to insufficient clinical data, and 1 patient was excluded due to discharge within 24 hours after admission. Finally, 34 tetanus patients were included as the study objects.” 

2．Comment:The study was conducted in adults; however, the age range of adults was not stated. This should be part of the inclusion criteria.

Reply: The article has added inclusion and exclusion criteria for research patients，and highlighted in red font.“Inclusion criteria were as follows: ① Patients clinically diagnosed with tetanus; ② Age>18 years old; ③ Complete clinical data. Exclusion criteria were: ① Discharge or death within 24 hours of admission; ② Incomplete clinical data.”

3．Comment:The study's retrospective design, which limits control over selection bias and missing data, should be considered a drawback in addition to those mentioned in the discussion section. The recommendation for a much larger study is in order; however, this preferably should be a prospective study.

Reply: In the discussion section, corresponding discussions have been added and highlighted in red font.“However, this study is retrospective in nature, which inherently entails drawbacks such as potential data bias, numerous confounding factors, recall bias, and difficulties in ensuring data quality and completeness. In future studies, we will conduct a multi-center, prospective study with an expanded sample size to validate and refine the findings of this research, thereby enhancing its clinical application value.”

4．Comment:The references need to be reviewed; most do not conform to PLOS One reference guideline (https://journals.plos.org/plosone/s/submission-guidelines). Furthermore, I could not find references 6, 15 and 16 from literature search.

Reply: 17 references were modified to meet the formatting requirements of the journal.References 5, 15, and 16 have been replaced and the format has been modified.

5．Comment:A majority of the References were more than five years; ideally, at least 60-70% of the References should be less than 5 years post publication. This must be addressed.

Reply: The references in the article have been revised. 17 references, all published in the past 5 years.

Reviewer #1: 

1-Comment:There are few scattered spelling mistakes like the word "conformed" in Page 9 and the word "tracheotomy" in Page 18.

Reply: Thank you to the reviewer for their meticulous and conscientious work. The author has checked the entire text and corrected any spelling errors

2-Comment:The author highlighted that the total sample size was 34. However, what was the sample size before the exclusion criteria, in other words, how many patients were admitted with tetanus?

Reply:In the results section, corresponding explanations have been added and highlighted in red font.“During the study period, a total of 41 tetanus patients were admitted to the Second People's Hospital of Fuyang. Among them, 6 patients were excluded due to insufficient clinical data, and 1 patient was excluded due to discharge within 24 hours after admission. Finally, 34 tetanus patients were included as the study objects.”

3-Comment:The method section seems a bit confusing and not reproducible. The author stated that the patients were categorised into ICU and non-ICU group. The word "categorised" seems inappropriate in this context as patients were not randomised. [This categorisation introduces bias]

Reply:To avoid misunderstandings, the author has made modifications to the sentence in the 1.2 data collection section and highlighted it in red font.“We collected clinical data from 34 tetanus patients and divided them into ICU and non ICU groups based on whether they received ICU treatment.”

4-Comment:The author need to illustrate when was the SII calculated. Was it calculated upon admission? Also, is there a role of monitoring serial or daily SII to predict progression of severity and/or morbidity?

Reply:（1）The SII is calculated within 24 hours of the patient's admission. This is explained in the 1.2 data collection section and highlighted in red font.（2）Thank you to the reviewer for providing a great research idea. Next, we will continuously monitor the SII of tetanus patients and study its guiding role in patient medical treatment.

5-Comment:Was the treatment similar to both arms of the study (the ICU and non-ICU groups)?

Reply:The author added a 1.3 treatment section in the 1 Materials and Methods section to introduce the treatment methods for the ICU and non ICU groups.

6-Comment:The discussion section highlights the importance of timely management. However, it does not answer the topic clearly which is the role of SII as this is the cornerstone of this manuscript and their is lack of literature review.

 Reply: In the discussion section, corresponding discussions have been added and highlighted in red font.“However, this study is retrospective in nature, which inherently entails drawbacks such as potential data bias, numerous confounding factors, recall bias, and difficulties in ensuring data quality and completeness. In future studies, we will conduct a multi-center, prospective study with an expanded sample size to validate and refine the findings of this research, thereby enhancing its clinical application value.”

Reviewer #2: 

Thank you to Professor Yahya Ali Abdulkareem Abodea for reviewing and guiding this article.The author has made modifications to the article, article format, and reference format to meet the publication requirements of the journal PLOS ONE, and has marked them in red in the manuscript.

The main revision in the paper and the response to the reviewers' comments are detailedly indicated in the attached file“Responses to comments”,and all changes are also highlighted in the manuscript.

We hope you will find our revised manuscript acceptable for publication.

Yours sincerely,

Huang Jizheng

---

## [Editor Report · Decision Letter 1]

9 Dec 2024

Systemic Immune Inflammation Index as a Predictor of Disease Severity in Tetanus Patients: A retrospective Observational Study

PONE-D-24-16126R1

Dear Dr. Huang Jizheng

We’re pleased to inform you that your manuscript has been judged scientifically suitable for publication and will be formally accepted for publication once it meets all outstanding technical requirements.

Kind regards,

Innocent Ijezie Chukwuonye, MBBS, FMCP(Internal Medicine)

Academic Editor

PLOS ONE
---

## [Editor Report · Acceptance letter]

19 Dec 2024

PONE-D-24-16126R1 

PLOS ONE

Dear Dr. Jizheng, 

I'm pleased to inform you that your manuscript has been deemed suitable for publication in PLOS ONE. Congratulations! Your manuscript is now being handed over to our production team.

Kind regards, 

on behalf of

Dr. Innocent Ijezie Chukwuonye 

Academic Editor

PLOS ONE